# Spatial Localization of Transformer Inspection Robot Based on Adaptive Denoising and SCOT-β Generalized Cross-Correlation

**DOI:** 10.3390/s24154937

**Published:** 2024-07-30

**Authors:** Hongxin Ji, Chao Zheng, Zijian Tang, Xinghua Liu, Liqing Liu

**Affiliations:** 1School of Electrical Engineering, China University of Mining and Technology, Xuzhou 221116, China; a1277808652@126.com (C.Z.); ts23230145p31@cumt.edu.cn (Z.T.); 2College of Mechanical and Electronic Engineering, Shandong Agricultural University, Tai’an 271018, China; lxh9357@163.com; 3State Grid Tianjin Electric Power Research Institute, Tianjin 300180, China; liulq328@126.com

**Keywords:** empirical mode decomposition, generalized cross correlation, robot localization, transformer, ultrasonic localization

## Abstract

In the detection process of the internal defects of large oil-immersed transformers, due to the huge size of large transformers and metal-enclosed structures, the positional localization of miniature inspection robots inside the transformer faces great difficulties. To address this problem, this paper proposes a three-dimensional positional localization method based on adaptive denoising and the SCOT weighting function with the addition of the exponent β (SCOT-β) generalized cross-correlation for L-type ultrasonic arrays of transformer internal inspection robots. Aiming at the strong noise interference in the field, the original signal is decomposed by an improved Empirical Mode Decomposition (EMD) method, and the optimal center frequency and bandwidth of each mode are adaptively searched. By extracting the modes in the frequency band of the positional localization signal, suppressing the modes in the noise frequency band, and reconstructing the Intrinsic Mode Function (IMF) of the independently selected superior modal components, a signal with a high signal-to-noise ratio is obtained. In addition, for the traditional mutual correlation algorithm with a large delay estimation error at a low signal-to-noise ratio, this paper adopts an improved generalized joint weighting function, SCOT-β, which improves the anti-jamming ability of the generalized mutual correlation method at a low signal-to-noise ratio by adding an exponential function to the denominator term of the SCOT weighting function’s generalized cross-correlation. Finally, the accurate positional localization of the transformer internal inspection robot is realized based on the quadratic L-array and search-based maximum likelihood estimation method. Simulation and experimental results show the following: the improved EMD denoising method better improves the signal-to-noise ratio of the positional localization signal with a lower distortion rate; in the transformer test tank, which is 120 cm in length, 100 cm in width, and 100 cm in height, based on the positional localization method in this paper, the average relative positional localization error of the transformer internal inspection robot in three-dimensional space is 2.27%, and the maximum positional localization error is less than 2 cm, which meets the requirements of engineering positional localization.

## 1. Introduction

Although the monitoring and diagnostic technology of power transformers has made significant progress, the metal sealing of oil-immersed transformers leads to poor internal visibility. Therefore, the search for internal defects and faults in oil-immersed transformers has always been a difficult problem plaguing substation maintenance personnel [1,2,3]. In order to accurately determine the internal defects of the transformer, manual drilling into the transformer or hanging cover is usually used for inspection. However, this method has three major problems: low efficiency, poor accuracy, and high risk [4,5].

With the progress of robotics, micro-robots for inspection inside the oil-immersed transformer can analyze and locate transformer faults more intuitively and quickly [6,7,8,9,10]. In the process of inspection inside the transformer, the micro-robotic fish needs to know its position information in the transformer oil, and then it can complete the path planning and inspection tasks according to the control instructions. Therefore, the localization of the micro-robotic fish in the transformer oil is the first and key issue for the completion of the inspection task. Due to the special environment constraints inside the transformer, there is a lack of accurate and effective localization methods [11,12,13]. Consequently, it is of great significance to study the localization method of the micro-robotic fish applicable to the special application environment inside the transformer.

Currently, positioning methods based on ultrasonic sensor arrays mainly include positioning technology based on time delay estimation, positioning technology based on high-resolution spectrum estimation, and positioning technology based on controllable beamforming. The technology of time delay estimation is mainly divided into two parts: time delay estimation and location estimation. Time delay estimation calculates the time difference between the sound source and the ultrasonic array sensor. Time delay estimation algorithms include generalized cross-correlation (GCC) [14,15,16] and least mean square (LMS) adaptive filtering [17,18]. Position estimation refers to the estimation of sound source coordinates through a spatial search using the time difference obtained by time delay estimation and the known geometric coordinates of ultrasonic array sensors. Time delay estimation is sensitive to reverberation and noise. The sound source location error is large at high reverberations or low signal-to-noise ratios.

Based on the high-resolution spectrum estimation technology, the sound source signals collected by the ultrasonic array sensor are arranged into a geometric shape, and then the angle and distance of the sound source are obtained according to the spatial spectrum of the sound source signals. The commonly used methods are as follows: the Auto-Regressive (AR) model, the Maximum Entropy (ME) method, and the eigenvalue decomposition method [19]. High-resolution spectrum estimation technology requires that the signal be narrowband, the sound source is located in the far field of space, the sound wave is a plane wave, and the ultrasonic array sensor has consistent characteristics. In addition, the high-resolution spectrum estimation technology has a large amount of computation involving matrix transformation and covariance matrix inversion. It is also vulnerable to reflection interference of signals. This degrades the positioning performance of the method.

Beamforming positioning technology processes the signals collected by ultrasonic array and forms a beam by weighted summation. Then it guides the beam by searching for the possible space of the sound source. The point with the largest output power is the location of the sound source [20]. The algorithm has a certain anti-interference ability. Presently, beamforming technology has been widely studied and applied, mainly focusing on the two-dimensional localization of sound sources. Traditional beamforming algorithms have low spatial resolution and a weak ability to suppress substantial noise interference. As the transformer inspection robot positioning in this paper is three-dimensional, and the internal space of the transformer is narrow, the on-site noise interference is substantial. Therefore, the traditional beamforming algorithm is difficult to apply directly to the accurate three-dimensional positioning of the transformer inspection robot.

Aiming at the problem of inaccurate robotic ultrasonic positional localization inside the transformer, this paper proposes a spatial positional localization method based on adaptive denoising and SCOT-β generalized inter-correlation. In terms of denoising, the IMF is obtained by decomposing the signal through an improved EMD method, adaptively searching for the optimal center frequency and bandwidth of each mode during the iterative updating process, and selecting the superior IMF and reconstructing it based on the correlation coefficient between each IMF and the original signal. For the time delay calculation, the improved generalized joint weighting function SCOT-β is used, which improves the anti-jamming ability of the generalized mutual correlation method at low signal-to-noise ratios by adding an exponential function to the denominator term of the SCOT weighting function. Finally, a quadratic L-shaped array and a search-based maximum likelihood estimation method are used for the acoustic source positional localization of the robot inside the transformer.

## 2. Ultrasonic Positional Localization Strategies for Transformer Internal Inspection Robots

### 2.1. Ultrasound Signal Denoising Based on Improved EMD Method

In the near-field model, ultrasound can be viewed as a point source, which propagates around it in a spherical form. Most of the propagation channels in the transformer are for transformer-insulating oil, which can only propagate longitudinal waves but not transverse waves. At 20 °C, the ultrasonic wave propagation speed in different media is shown in Table 1, and the default ultrasonic propagation speed is set to 1400 m/s for experimental and field testing.

The noise at the experimental site is large and complex and has a large impact on the time delay estimation of the weak effective signal components. Therefore, it is necessary to denoise the original signal. Since the EMD method is an adaptive decomposition method, which does not need to make any assumptions about the characteristics of the signal, the decomposed eigenmode function components are strictly orthogonal, and the signal characterization can be carried out in different frequency bands with a strong frequency resolution, and the EMD-based denoising method can be used.

The EMD decomposition method needs to follow two principles:The number of over-zero points of the IMF component should be the same as or differ only by one from the number of extreme points;The upper and lower envelopes have an average value of 0 at any point in the entire signal.

The EMD method is calculated as follows:Find all the extreme points of the signal x(t) (endpoints are processed by mirroring);The envelope of the upper and lower extreme points *e_max_(t)* and *e_min_(t)* are fitted with three time spline curves, and the mean value of the upper and lower envelopes *m(t)* is found. *h(t)* can be expressed as *h(t)* = *x(t)* − *m(t)*; the mean value is expressed as follows:
(1)$$m(t)=\frac{\left[e_{\max}(t)+e_{\min}(t)\right]}{2}$$

3.Determine whether h(t) is an IMF based on a predefined criterion;4.If not, replace *x(t)* with h(t) and repeat the above steps until h(t) satisfies the criterion, then h(t) is the *IMF_i_(t)* that needs to be extracted;5.For each order of IMF obtained, subtract it from the original signal. Repeat the above steps until the last remaining part of the signal r_n_ is a monotonic sequence or a constant value sequence.

Thus, the EMD method decomposes the original signal *x(t)* into a series of IMFs and a linear superposition of the remaining parts:(2)$$x(t)=\sum_{i=0}^{N}IMF_{i}(t)+r_{n}(t)$$

After the decomposition is completed, the appropriate IMF is then selected for reconstruction based on the degree of correlation between the modal function and the original positional localization signal.

EMD is an empirical-based algorithm, which has defects such as the endpoint effect and modal aliasing. Aiming at problems such as the modal aliasing that may occur in the signal decomposition of the EMD method, Dragomiretskiy proposed an improved EMD method, which is based on classical Wiener filtering. By solving the variational problem, the optimal center frequency and bandwidth of each mode are searched adaptively during the iterative updating process, the modes in the desired frequency band are extracted, and the modes in the noise band are suppressed. Ultimately, signal denoising can be realized by reconstructing the signal [21,22]. This method has a more solid theoretical foundation to reduce the non-smoothness of time-series signals with high complexity and strong nonlinearity and is suitable for non-smooth time-domain signals.

The solution process of the improved EMD method contains two main constraints:The sum of the bandwidths of the center frequencies of each modal component is required to be a minimum;The sum of all modal components is equal to the original signal.

The improved EMD method redefines the intrinsic evidence modal function for a finite bandwidth with more stringent constraints, and its modal components are defined as the component modal functions for AM and FM with mathematical expressions:(3)$$u_{k}(t)=A_{k}(t)\cos(\phi_{k}(t))$$
where $A_{k}(t)$ is the envelope amplitude of the signal $u_{k}(t)$, and $\varphi_{k}(t)$ is the instantaneous phase.

The constrained variational modeling is as follows:(4)$$\left\{\begin{array}{l} \underset{\left\{u_{k}\},\{\omega_{k}\right\}}{\min}\{\sum_{k}||\partial_{t}[(\delta (t)+\frac{j}{\pi t})*u_{k}(t)]e^{-j\omega_{k}t}|{|}_{2}^{2}\}\\ s.t.\sum_{k=1}^{K}u_{k}=f \end{array}\right.$$
where f is the original signal, K denotes the number of modes of the signal decomposition, $\left\{u_{k}\}=\{u_{1},u_{2},\ldots ,u_{k}\right\}$,$\left\{\omega_{k}\}=\{\omega_{1},\omega_{2},\ldots ,\omega_{k}\right\}$ are all the modal components and their corresponding center frequencies, respectively, * is the convolution operator, and δ(t) is the impulse function.

To solve the above constrained variational problem, a quadratic penalty term α and a Lagrange multiplier λ are introduced to transform the constrained variational problem into an unconstrained variational problem. The preset value of K determines the number of IMF components, and the penalty coefficient α determines the bandwidth of the IMF components. The smaller the penalty coefficient is, the larger the bandwidth of each IMF component is; too large a bandwidth will cause some components to contain signals from other components; the larger the value of α is, the smaller the bandwidth of each IMF component is, and too small a bandwidth will cause some signals to be lost in the decomposed signal. This coefficient commonly takes values in the range of 1000 to 3000. In this paper, parameters were optimized by Grey Wolf Optimization (GWO), and we approximate that α is 2000 and K is 6. The resulting augmented Lagrangian expression is obtained as follows:(5)$$\begin{array}{l} L\{\{u_{k}\},\{\omega_{k}\},\lambda \}=\alpha \sum_{k}||\partial_{t}[(\delta (t)+\frac{j}{\pi t})*u_{k}(t)]e^{-j\omega_{k}t}{\left|\right|}_{2}^{2}\\ +||f(t)-\sum_{k}u_{k}(t){\left|\right|}_{2}^{2}+\langle \lambda (t),f(t)-\sum_{k}u_{k}(t)\rangle \end{array}$$

Solving for each mode and its center frequency, the following results are obtained:(6)$${\hat{u}}_{k}^{n+1}(\omega )=\frac{\hat{f}(\omega )-\sum_{i\neq k}{\hat{u}}_{i}(\omega )+\frac{\hat{\lambda }(\omega )}{2}}{1+2\alpha {\left(\omega -\omega_{k}\right)}^{2}}$$
(7)$$\omega_{k}^{n+1}=\frac{\int_{0}^{\infty }\omega |{\hat{u}}_{k}(\omega ){|}^{2}d\omega }{\int_{0}^{\infty }|{\hat{u}}_{k}(\omega ){|}^{2}d\omega }$$
(8)$${\hat{\lambda }}_{k}^{n+1}={\hat{\lambda }}^{n}(\omega )+\gamma (\hat{f}(\omega )-\sum_{k}{\hat{u}}_{k}^{n+1}(\omega ))$$
where *n* represents the number of iterations, ω represents the frequency value, ${\hat{u^{n+1}}}_{k}(\omega )$, $\hat{f}(\omega )$, and $\hat{\lambda }(\omega )$ are the Fourier transforms of ${u^{n+1}}_{k}(\omega )$, f(ω), and λ(ω), respectively, and *γ* denotes the noise tolerance.

The IMF is obtained by performing a Fourier inverse transform of the desired modal *uˆ_k_(ω)*. After the decomposition is completed, by calculating the correlation coefficient of each IMF, the denoised result can be obtained by taking the appropriate IMF reconstruction.

Improved EMDs offer unique advantages over traditional EMDs:The number of modes to be obtained can be specified;The decomposed IMFs all have independent center frequencies and exhibit sparsity characteristics in the frequency domain, possessing the qualities of sparse study;In the process of solving the IMF, the endpoint effect that occurs in the ordinary EMD decomposition is avoided by way of mirror extension;Effective avoidance of modal aliasing, with appropriate selection of K values.

### 2.2. Ultrasonic Positional Localization Method Based on an Improved Generalized Mutual Correlation Algorithm

In order to carry out three-dimensional spatial ultrasonic positional localization for the transformer internal inspection robot, it is necessary to set up an ultrasonic array inside the transformer to receive the ultrasonic signals issued by the robot, which is represented by s(t), which will be affected by the surrounding noise, reverberation, etc. Assuming that M_1_ and M_2_ are the two ultrasonic signal-receiving sensors inside the transformer, and based on the theory of signal propagation, the signals received by them are, respectively, as follows:(9)$$x_{1}(t)=s(t-\tau_{1})+n_{1}(t)$$
(10)$$x_{2}(t)=s(t-\tau_{2})+n_{2}(t)$$
where s(t) is the ultrasound signal from the transformer internal inspection robot, $n_{1}(t)$ and $n_{2}(t)$ are the Gaussian white noise of the two ultrasound positional localization receivers, and $\tau_{1}$ and $\tau_{2}$ are the times for the sound source to reach the receivers M_1_ and M_2_, respectively. Then, $\tau_{12}=\tau_{1}-\tau_{2}$ is the time delay for the ultrasound signal to reach M_1_ and M_2_.

The basic mutual correlation algorithm is computationally small and easy to implement. The model *s*(*t*), *n*_1_(*t*), *n*_2_(*t*) are mutually uncorrelated smooth stochastic varying processes, then the mutual correlation function of *x_1_(t)* and *x_2_(t)* can be expressed as follows:(11)$$R_{x_{1}x_{2}}(\tau )=E[x_{1}(t)x_{2}(t-\tau )]$$

According to Wiener Sinchen’s theorem, the mutual correlation function and its reciprocal power spectral density are Fourier transform pairs of each other; then the mutual correlation function of *x_1_(t)* and *x_2_(t)* can be expressed as follows:(12)$$R_{x_{1}x_{2}}(\tau )=\int_{-\infty }^{+\infty }G_{x_{1}x_{2}}(\omega )e^{-j\omega \tau }d\omega =\int_{-\infty }^{+\infty }X_{1}(\omega )X_{2}^{*}(\omega )e^{-j\omega \tau }d\omega$$
where $X_{1}(\omega )$ and $X_{2}(\omega )$ are the Fourier transforms of $x_{1}(t)$ and $x_{2}(t)$, respectively, and $G_{x_{1}x_{2}}(\omega )$ is referred to as the reciprocal power spectra of the ultrasound signals $x_{1}(t)$ and $x_{2}(t)$, and the transverse coordinate at the corresponding maximal peak value of $R_{x_{1}x_{2}}(\tau )$ is the required delay estimation, as shown in Equation (13).
(13)$$\tau_{x_{1}x_{2}}=\arg_{\tau }\max R_{x_{1}x_{2}}(\tau )$$

Although the cross-correlation algorithm has the advantages of simplicity and fast operation, the performance is often affected by a variety of factors, such as signal autocorrelation, reverberation, and so on. In practice, the peaks of the correlation function are often affected by noise, making the extreme peaks spread out and become inconspicuous, which makes it difficult to accurately find the location of the extreme points. When there is serious noise and reverberation, the basic correlation function may also appear with multiple peaks, further aggravating the difficulty of detecting the actual peak.

In order to overcome the shortcomings of the basic cross-correlation algorithm in the case of a low signal-to-noise ratio with a large delay estimation error, firstly, the fast Fourier transform is performed on the two received signals, respectively, to obtain the frequency domain form of the cross-correlation function of the two signals; then the cross-correlation function is weighted in the frequency domain, and the results of the weighted computation are fast Fourier inverted and converted to the time-domain form, and then peak-detection is carried out. As shown in Figure 1. The generalized cross-correlation function sharpens the peak of *R_x1x2_(τ)* at the time delay, thus improving the delay estimation performance. The generalized cross-correlation function *R_x1x2_(τ)* is as follows:(14)$$R_{x_{1}x_{2}}(\tau )=\int_{-\infty }^{+\infty }\varphi_{12}(\omega )X_{1}(\omega )X_{2}^{*}(\omega )e^{-j\omega \tau }d\omega$$
where $X_{1}(\omega )$ and $X_{2}(\omega )$ are the Fourier transforms of $x_{1}(t)$ and $x_{2}(t)$, respectively, while $G_{x_{1}x_{2}}(\omega )$ is referred to as the reciprocal power spectrum of the ultrasound signals $x_{1}(t)$ and $x_{2}(t)$, and $\varphi_{12}$ is the joint weighting function.

In practice, there are different choices depending on the noise and reverberation, so that more pronounced peaks can be obtained. However, choosing the appropriate weighting function is a difficult task due to the finite window length and low signal-to-noise ratio. To cope with the effect of the signal strength change on TDOA, a better solution is to pre-whiten the positional localization signal before obtaining its mutual spectrum. At this time, the weighting function is set as follows:(15)$$\varphi_{12}(\omega )=\frac{1}{\sqrt{G_{x_{1}x_{1}}(\omega )G_{x_{2}x_{2}}(\omega )}}$$

This is known as the Smoothed Coherence Transform (SCOT) method. This weighting function takes into account both received signals and overcomes the effect of the peak broadening of the correlation function, which produces spurious peaks. However, when the power spectral densities of the two signals are equal, it will broaden the peaks of the correlation function and produce spurious peaks leading to false estimations. The delay estimation performance of the SCOT algorithm will vary with the SNR. The SCOT method is theoretically superior to the generalized cross-correlation method, but this superiority holds only at low noise levels.

At high signal-to-noise ratios, the peak of the generalized cross-correlation algorithm based on the SCOT weighting function is more pronounced, and the ultrasound signal time delay can be effectively calculated. However, the performance of the algorithm decreases sharply at low signal-to-noise ratios. In order to improve the performance of this algorithm in the case of a low signal-to-noise ratio, and improve its stability and accuracy, this paper proposes an improved generalized joint weighting function. Considering that the weighting factor is affected by the two channels as well as the smaller energy of the sampled signal in the small signal-to-noise ratio environment, combined with the advantages and disadvantages of the SCOT weighting function, this paper introduces a new weighting factor, denoted as SCOT-β, and the expression of the weighting function is as follows:(16)$${\left(\frac{1}{\sqrt{G_{x_{1}x_{1}}*G_{x_{2}x_{2}}}}\right)}^{\beta }$$

By adding an exponential function to the denominator term of the SCOT weighting function, the generalized cross-correlation algorithm’s anti-interference ability at low signal-to-noise ratios is improved. Simulation experimental results show that this weighting function can sharpen the generalized cross-correlation peaks, ensure the accuracy of delay estimation, and improve the stability of the generalized cross-correlation algorithm in low signal-to-noise ratio cases.

### 2.3. Three-Dimensional Spatial Positional Localization Algorithms for Transformer Internal Inspection Robots

To determine the three-dimensional coordinates of a sound source in space, four array elements are theoretically required. The coordinates of the sound source obtained by the planar four-element array positional localization are two solutions in the upper and lower spatial domains bounded by the array plane, and it is necessary to choose the appropriate solution of the two according to the actual situation.

In order to meet the requirements of the system for positioning accuracy and reduce the computational complexity of the system and the difficulty of implementation, this paper adopts a two-dimensional planar array structure, designed with a four-element L-type planar ultrasonic array structure. In the spatial right-angled coordinate system shown in Figure 2, the ultrasonic array consists of four omnidirectional waterproof piezoelectric ceramic ultrasonic probes, containing three ultrasonic probes in the X-axis direction and two ultrasonic probes in the Y-axis direction, which constitutes an L-type. the coordinates of the four ultrasonic probes are M_0_ (0,0,0), M_1_ (d,0,0), M_2_ (2d,0,0), and M_3_ (0,d,0), and the coordinates of the acoustic source are set as S (x_s_, y_s_, z_s_).

The distance from the sound source to the positional localization array of the transformer internal inspection robot can be obtained by calculating the product of the sound velocity and time delay as shown in Equation (17).
(17)$$\left\{\begin{array}{l} x_{s}^{2}+y_{s}^{2}+z_{s}^{2}=c^{2}{\left(t_{1}-t_{0}\right)}^{2}\\ {\left(x_{s}-d\right)}^{2}+y_{s}^{2}+z_{s}^{2}=c^{2}{\left(t_{2}-t_{0}\right)}^{2}\\ {\left(x_{s}-2d\right)}^{2}+y_{s}^{2}+z_{s}^{2}=c^{2}{\left(t_{3}-t_{0}\right)}^{2}\\ x_{s}^{2}+{\left(y_{s}-d\right)}^{2}+z_{s}^{2}=c^{2}{\left(t_{4}-t_{0}\right)}^{2} \end{array}\right.$$
where $t_{0}$ is the time when the sound source emits the signal, $t_{1}$, $t_{2}$, $t_{3}$, $t_{4}$ are the times when the signals are received by different ultrasound probes, $\tau_{10}=t_{2}-t_{1}$, $\tau_{20}=t_{3}-t_{1}$, $\tau_{30}=t_{4}-t_{1}$ are the time delays of the signals received by the ultrasound probes and the reference probes, respectively, c is the speed of sound, and d is the distance between the ultrasound probes.

According to the above equation, the solution to obtain the 3D coordinate position of the sound source of the inspection robot is as follows:(18)$$\left\{\begin{array}{l} x_{s}=\frac{2cq\tau_{10}+c^{2}\tau_{10}^{2}-d^{2}}{-2d}\\ y_{s}=\frac{2cq\tau_{30}+c^{2}\tau_{30}^{2}-d^{2}}{-2d}\\ z_{s}=\pm \sqrt{q^{2}-x_{s}^{2}-y_{s}^{2}} \end{array}\right.$$
where
(19)$$q=[2d^{2}-c^{2}(\tau_{20}^{2}-2\tau_{10}^{2})]/2c(\tau_{20}-2\tau_{10})$$

There is a large noise interference in the experimental site, which makes the time delay estimation results have a large error. As the geometric positional localization method is more sensitive to the time delay difference, it is easy to reduce the accuracy of the final positional localization results. Therefore, this paper adopts the search-based maximum likelihood estimation method to carry out the three-dimensional positional localization of the transformer internal inspection robot.

For a positional localization array consisting of M ultrasonic transducers, let the spatial coordinates of the ith transducer m_i_(i = 1, 2,..., M) be r_i_ = [x,y,z]^T^, and let the spatial coordinates of the acoustic source r_s_ = [x_s_,y_s_,z_s_]^T^. Let m_0_ be the reference origin; then the distance from mi to m_0_ is as follows:(20)$$\begin{array}{l} r_{i}=||r_{i}||=\sqrt{x_{i}^{2}+y_{i}^{2}+z_{i}^{2}}\\ i=1,\ldots ,M \end{array}$$

The distances at which the sound source reaches m_0_ and m_i_ are, respectively, as follows:(21)$$r_{s}=||r_{s}||=\sqrt{x_{s}^{2}+y_{s}^{2}+z_{s}^{2}}$$
(22)$$d_{i}=||r_{i}-r_{s}||$$

The distance difference between the sound source arriving at m_i_ and m_j_ is obtained as follows:(23)$$\begin{array}{l} \delta_{ij}=d_{i}-d_{j}\\ i,j=1,2,\ldots ,M,i\neq j \end{array}$$

In addition, after obtaining an estimate of the time difference between the signals acquired by the two sensors, the estimate of the distance difference between m_i_ and m_j_ can be expressed as follows:(24)$${\hat{\delta }}_{ij}=c*{\hat{\tau }}_{ij}$$

Up to this point, the TDOA-based sound source positional localization problem can be summarized as follows: estimating the source location *r_s_* by minimizing the error values of Equations (23) and (24) when given the estimated values of *r_i_* and *r_j_* as well as the time delays of both.

In an array of M sensors, any one-to-one combination between sensors produces M (M − 1)/2 pairs, resulting in M (M − 1)/2 time delays t_ij_, (i,j = 0, 1,..., M−1 and I ≠ j). In the absence of noise, the space produced by these estimates is M−1 dimensional. Without loss of generality, δ_i0_, (i = 1, 2,..., M − 1) is chosen as a set of bases for the space.

In an ideal situation,
(25)$${\hat{\delta }}_{i0}=\delta_{i0}=||r_{i}-r_{s}||-||r_{s}||$$

Because of the presence of interference (assumed to be additive noise) in the environment, the estimate of the distance difference in the realistic case should be as follows:(26)$$\begin{array}{l} {\hat{\delta }}_{i0}=g_{i}(r_{s})+\varepsilon_{i}\\ i=1,2,\ldots ,M-1 \end{array}$$
where
(27)$$g_{i}(r_{s})=||r_{i}-r_{s}||-||r_{s}||$$

In Equation (26), ε_i_ is the measurement error. Equation (26) can be written in vector form as follows:(28)$$\hat{\delta }=g(r_{s})+\varepsilon$$
where
(29)$$\begin{array}{l} \hat{\delta }={\left[{\hat{\delta }}_{10},{\hat{\delta }}_{20},{\hat{\delta }}_{30},\ldots ,{\hat{\delta }}_{(M-1)0}\right]}^{T}\\ g(r_{s})={\left[g_{1}(r_{s}),g_{2}(r_{s}),\ldots ,g_{M-1}(r_{s})\right]}^{T}\\ \varepsilon ={\left[\varepsilon_{1},\varepsilon_{2},\ldots ,\varepsilon_{M-1}\right]}^{T} \end{array}$$

By making specific assumptions about the distribution of the error ε, the estimated parameters are made to be optimal in a probabilistic sense, which is the maximum likelihood estimation method.

Assuming that the source vector position is s, given N sets of ultrasonic receiving transducers with vector positions m_i1_, m_i2_, with *i* ∈ (1,N), each transducer pair has an estimated delay, while its true delay is as follows:(30)$$T(\{m_{i1},m_{i2}\},s)=\frac{\left\|m_{i1}-s\right\|-\left\|m_{i2}-s\right\|}{c}$$

In practice, the effect of noise can lead to errors between the true and estimated values. There is an estimation variance σ_i_^2^, which is related to factors such as the signal-to-noise ratio, the site reverberation, and delay estimation algorithms.

The maximum likelihood estimation DML can be expressed as follows:(31)$${\hat{D}}_{ML}=\underset{s}{\arg\min(J_{ML}(s))}$$
where
(32)$$J_{ML}(s)=\sum_{i=1}^{N}\frac{1}{\sigma_{i}^{2}}{\left[\tau_{i}-T(\{m_{i1},m_{i2}\},s)\right]}^{2}$$

The maximum likelihood estimation method yields a more accurate position estimate, but T({m_i1_,m_i2_},s) is a nonlinear equation for s. Therefore, Equation (31) cannot be solved in a closed manner, and the position estimation involves a large number of searches in the possible positional localization space as a means of determining the location of the sound source. Although the target space can be efficiently reduced to the desired range by a fast-converging search method, this global search method is not very feasible due to computational limitations. This is especially true in real-time environments, where higher update rates or multiple sensors are required.

Therefore, it is necessary to analyze the time-delay characteristics to quickly narrow the positional localization range. In order to realize the positional localization more accurately, the spatial search method adopted in this paper is divided into two steps: Firstly, according to the result of the time delay estimation, the quadrant where the sound source is located is judged, so as to narrow down the search range of the sound source location. Then, an L-shaped array consisting of four ultrasound-receiving transducers is utilized to do the precise positional localization. Since there is no need to search the whole space, the computational amount is reduced, which also helps to analyze the positional localization solution to some extent, eliminating the fuzzy solution and improving the positional localization success rate.

## 3. Ultrasonic Spatial Positional Localization Simulation Test for Inspection Robots

### 3.1. Simulation Verification of Improved Adaptive Denoising Algorithm

During the validation process, the emitted ultrasound signal consisted of three sinusoidal signals with a frequency of 180 kHz. The sampling frequency is 2.5 MSa/s, and the number of sampling points is 4096. In order to simulate the effect of noise on the signals, Gaussian white noise with SNR = −5 dB was added to the signals, as shown in Figure 3.

The spectrum of the ultrasonic signal is concentrated around the 180 kHz frequency, as shown in Figure 4. In this paper, the ultrasonic signal with noise is decomposed by EMD and improved EMD, respectively, and the decomposed intrinsic modal components are shown in Figure 5 and Figure 6.

The results were reconstructed according to the correlation coefficient, as shown in Figure 7 and Figure 8.

In order to compare and evaluate the noise reduction performance of the improved EMD denoising algorithm and the traditional EMD denoising algorithm on the signal, this paper adopts the signal-to-noise ratio (SNR), the Root-Mean-Square Error (RMSE), and the Normalized Correlation Coefficient (NCC) as the evaluation indexes, in which the SNR is defined as follows:(33)$$SNR=10\mathrm{lg}\frac{\sum_{i=1}^{n}x^{2}(i)}{\sum_{i=1}^{n}{\left(x(i)-\overset{-}{x}(i)\right)}^{2}}$$

The RMSE is defined as follows:(34)$$RMSE=\sqrt{\frac{1}{n}\sum_{i=1}^{n}{\left(x(i)-\overset{-}{x}(i)\right)}^{2}}$$

The NCC is defined as follows:(35)$$NCC=\frac{\sum_{i=1}^{n}x(i)\cdot \overset{-}{x}(i)}{\sqrt{(\sum_{i=1}^{n}x^{2}(i))\cdot (\sum_{i=1}^{n}\overset{-}{x^{2}}(i))}}$$
where x(i) denotes the original signal, and $\overset{-}{x}(i)$ denotes the denoised signal.

The results of the noise reduction performance of different denoising methods are shown in Table 2. As shown in Table 2, compared with the traditional EMD denoising method and the bandpass filtering method, the improved EMD denoising method proposed in this paper has the largest SNR value, which indicates that the smaller the proportion of noise in the signal and the better the denoising effect, the smaller the RMSE, which indicates that the signal distortion is smaller and the NCC value is closer to 1, which indicates that the denoising effect is better.

### 3.2. Simulation Validation of Improved Generalized Cross-Correlation Delay Estimation Methods

Figure 9 shows the generalized cross-correlation results for GCC, SCOT, and SCOT-β for β taken as 0.2, 0.4, 0.6, and 0.8, respectively, at an SNR of −5 dB.

Comparing the generalized cross-correlation results of the three methods, the peaks of the curves obtained by SCOT-β are sharper and more obvious than those obtained by SCOT and GCC, and the measured delay estimation results are far more obvious than those of the other two methods, which indicates that the delay estimation performance of SCOT-β is better than that of GCC and SCOT.

Comparing the results when β is taken at different values, we find that the interference peak gradually becomes larger as the value of β increases. This indicates that the increase in the value of β taken reduces the delay estimation performance of SCOT-β to some extent. The generalized inter-correlation results are superior when β is taken as 0.2.

## 4. Ultrasonic Spatial Positional Localization Practical Test for Inspection Robots

### 4.1. Test Platform for Three-Dimensional Spatial Positional Localization

In order to verify the practicality of the proposed 3D spatial positional localization method, this paper built an experimental test platform, which mainly includes a transformer internal inspection robot, an ultrasonic positional localization array, data acquisition, a control platform, and other modules, as shown in Figure 10.

The oil tank used for the test platform is 1200 mm long, 1000 mm wide, and 1000 mm high. The oil tank is filled with 25 # transformer oil. The transformer internal inspection robot and ultrasonic array are all in the transformer oil. The overall structure of the transformer internal inspection robot is shown in Figure 11. Inspection robot mainly includes: #1-robot body shell, #2-ultrasonic emission sensor, #3-visual device, #4-infrared ranging module, #5-robot vertical propeller propulsion device, #6-robot horizontal propeller propulsion device, #7-pressure sensor, and robot control system.

The ultrasonic array mounted inside the transformer consists of four omnidirectional waterproof piezoelectric ceramic ultrasonic probes, with one ultrasound probe at the reference origin, two ultrasound probes in the horizontal direction, and one ultrasound probe in the vertical direction, in the form of an L-shaped array. In this experiment, four sensors were selected, as shown in Figure 12. The nominal frequency of the ultrasound probes is 180 kHz, and the coordinates of the four probes are M_0_ (0 cm, 0 cm, 0 cm), M_1_ (10 cm, 0 cm, 0 cm), M_2_ (20 cm, 0 cm, 0 cm), and M_3_ (0 cm, 10 cm, 0 cm), respectively. Let the coordinates of the sound source be S (x_s_ cm, y_s_ cm, z_s_ cm), the distance to the origin of the right-angle coordinate system be l_0_, and the distances between the sound source S and the four formants M_0_, M_1_, M_2_, and M_3_ be l_0_, l_1_, l_2_, and l_3_, respectively.

### 4.2. Data Acquisition

In the test, the inspection robot is moved into position (30 cm, 20 cm, −72 cm) by a three-dimensional spatial coordinate calibration device. The ultrasonic sensor on the top of the transformer internal inspection robot sends out a set of three consecutive sinusoidal ultrasonic signals with a frequency of 180 kHz every 500 ms. The ultrasonic array synchronously collects the ultrasonic signals, and the parameters of the data acquisition device are set as follows: the sampling rate of each channel is 2.5 MSa/s, the acquisition depth is 4096 points, and the triggering mode adopts the rising edge triggering. A set of ultrasound signals was acquired as shown in Figure 13.

The improved EMD denoising method was used to denoise the data separately, and the waveforms in Figure 14 are the results of denoising the original waveforms in Figure 13.

The denoised multigroup signals are substituted into the SCOT-β to calculate the delay results, as shown in Table 3.

### 4.3. Analysis of Positioning Test Results

The positional localization results can be obtained by the objective function search method, as shown in Table 4. According to the repeated positional localization results of the transformer internal inspection robot, the 3D spatial positional localization method of the inspection robot has a maximum positional localization error of 2 cm in the x-axis and an average positional localization error of 0.90 cm in the x-axis, a maximum positional localization error of 2 cm in the y-axis and an average positional localization error of 1.10 cm in the y-axis, and a maximum positional localization error of 2 cm in the z-axis and an average positional localization error of 1.15 cm in the z-axis. The maximum relative positioning error of the transformer internal inspection robot is 3.04%, and the average relative positioning error is 2.27%, and the positioning accuracy meets the engineering requirements.

## 5. Conclusions

Because it is difficult to use conventional LIDAR, vision cameras, the Global Position System (GPS), and other methods for transformer internal inspection robot positional localization, this paper proposed a new method of positional localization based on improved EMD decomposition, SCOT-β, and quaternary L-type ultrasonic arrays in an integrated manner.

Improved EMD decomposition and correlation coefficient-based denoising methods are used for signal denoising. Compared with the traditional EMD method, the SNR is improved from 3.62 dB to 6.59 dB, the NCC is improved from 81.37% to 89.66%, and the RMSE is reduced from 0.1009 to 0.0708; therefore, it can be obtained that the improved EMD denoising is better than the traditional EMD method for denoising high noise signals.

Compared with SCOT and GCC, the curve peaks obtained by SCOT-β proposed in this paper are sharper and more obvious, and the measured delay estimation accuracy is far better than the other two methods, which effectively realizes the sharpening of the peaks of the cross-correlation results, enhances the anti-jamming property of the generalized cross-correlation delay estimation, and improves the accuracy of the positional localization. The interference peak of the SCOT-β joint weighting function becomes larger as the value of β increases, indicating that the increase in the value of β reduces the delay estimation performance of SCOT-β; when β is taken as 0.2, the results of the generalized cross-correlation are relatively better.

After the signal denoising and the generalized mutual correlation method to find the time delay, this paper carries out the positional localization of the ultrasonic sound source by the search-based maximum likelihood estimation method. Several sets of experimental data are obtained with a step size of 1 cm, and the final positional localization of the sound source is obtained after processing: x_s_ = 30.90 cm, y_s_ = 18.90 cm, z_s_ = −70.85 cm. Compared with the actual position of the ultrasonic sound source, the absolute errors are as follows: Δx_s_ = 0.9 cm, Δy_s_ = 1.10 cm, Δz_s_ = 1.15 cm; and the relative errors are as follows: ζx_s_ = 3.0%, ζy_s_ = 5.5%, ζz_s_ = 1.6%. Therefore, the positioning accuracy meets the demand of robot positioning inside the transformer.

On the basis of simulation research, this paper builds a test platform for the three-dimensional spatial positional localization effect of a transformer internal inspection robot. By comparing the positional localization results of the inspection robot sound source at different positions, the effectiveness of the method proposed in this paper is verified, and the positional localization results show that the three-dimensional spatial average relative positioning error of the transformer internal inspection robot is 2.27%, and the maximum positional localization error is less than 2 cm, and the repeat positional localization error at different positions is small, which can meet the current engineering requirements of the inspection robot positional localization.

## Figures and Tables

**Figure 1 sensors-24-04937-f001:**
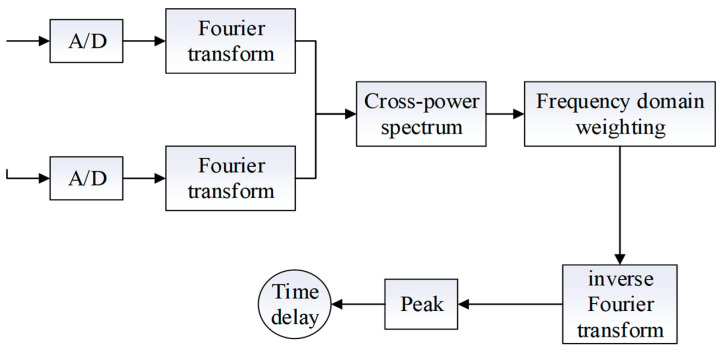
Schematic diagram of the generalized cross-correlation.

**Figure 2 sensors-24-04937-f002:**
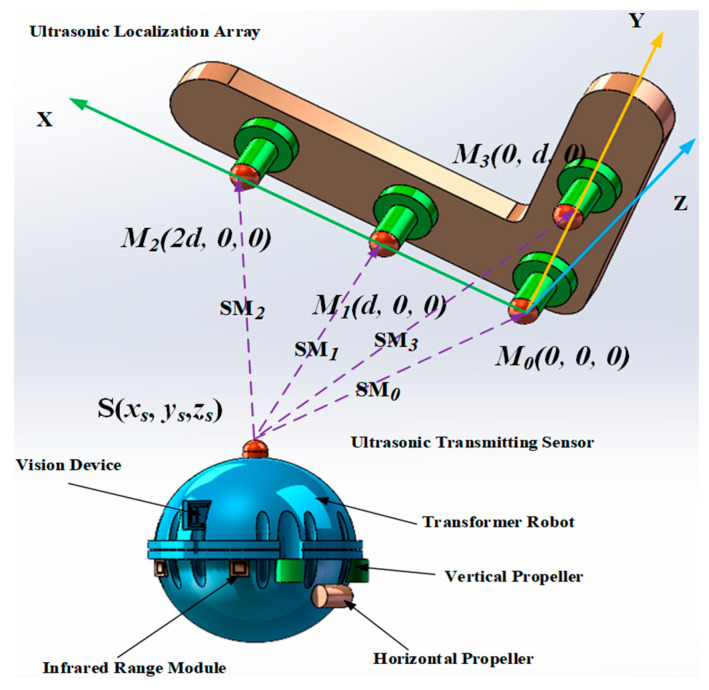
Schematic diagram of spatial positional localization with the quadratic ultrasound array.

**Figure 3 sensors-24-04937-f003:**
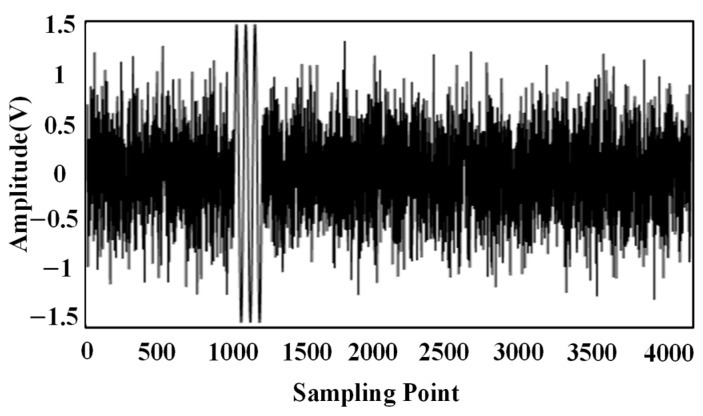
Simulated signal with a signal-to-noise ratio of −5 dB.

**Figure 4 sensors-24-04937-f004:**
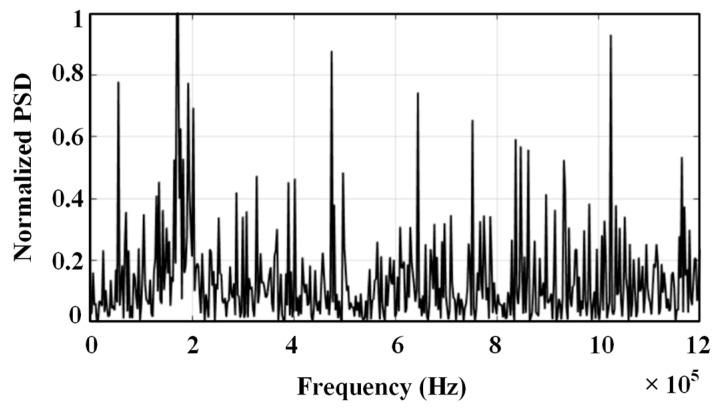
Spectrogram of the simulated ultrasound signal.

**Figure 5 sensors-24-04937-f005:**
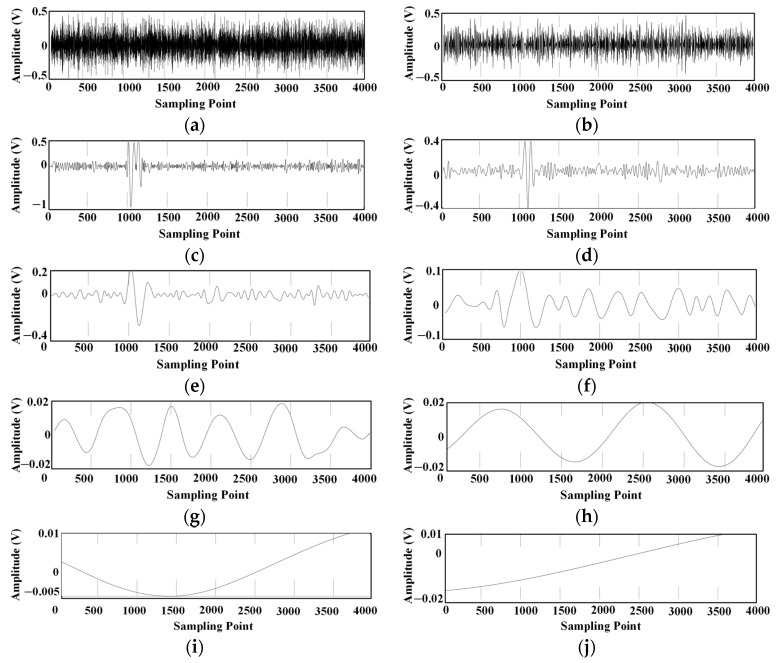
Intrinsic modal components obtained with EMD decomposition: (**a**) IMF1; (**b**) IMF2; (**c**) IMF3; (**d**) IMF4; (**e**) IMF5; (**f**) IMF6; (**g**) IMF7; (**h**) IMF8; (**i**) IMF9; (**j**) Residual.

**Figure 6 sensors-24-04937-f006:**
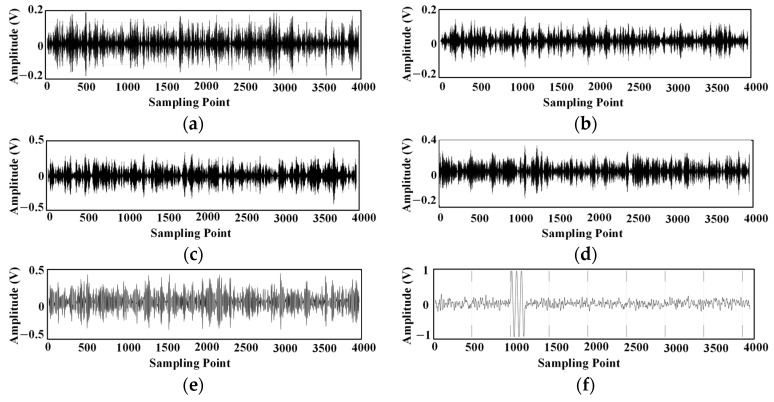
Intrinsic modal components obtained by the improved EMD algorithm decomposition: (**a**) IMF1; (**b**) IMF2; (**c**) IMF3; (**d**) IMF4; (**e**) IMF5; (**f**) IMF6.

**Figure 7 sensors-24-04937-f007:**
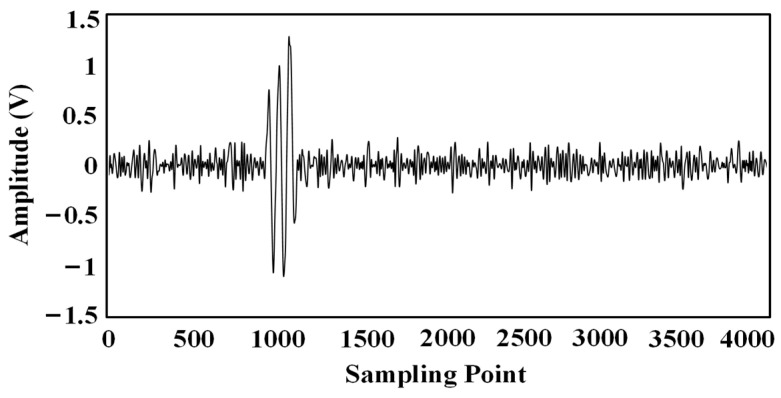
The denoising result with the EMD algorithm.

**Figure 8 sensors-24-04937-f008:**
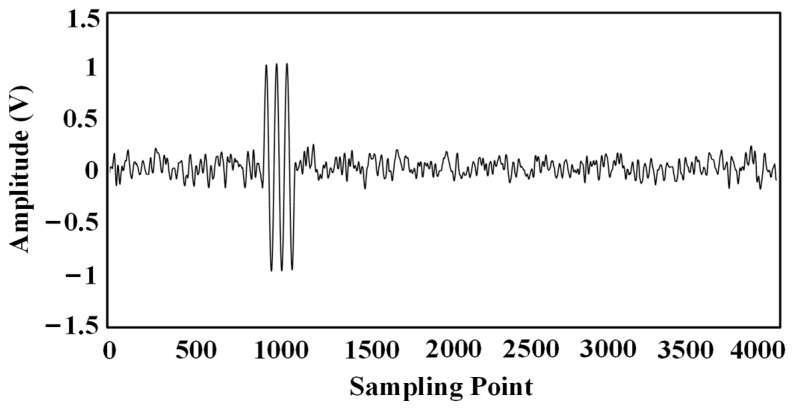
The denoising results of the improved EMD algorithm.

**Figure 9 sensors-24-04937-f009:**
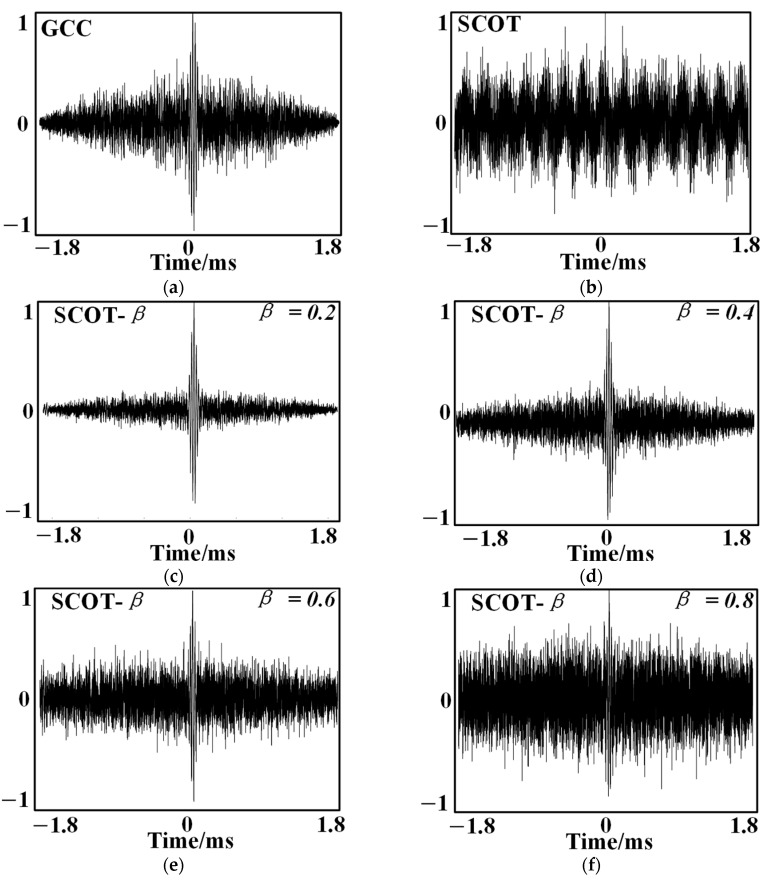
Cross-correlation results obtained with different methods: (**a**) the result of basic cross-correlation; (**b**) the result of SCOT generalized cross-correlation; (**c**) the result of SCOT-β (β = 0.2); (**d**) the result of SCOT-β (β = 0.4); (**e**) the result of SCOT-β (β = 0.6); (**f**) the result of SCOT-β (β = 0.8).

**Figure 10 sensors-24-04937-f010:**
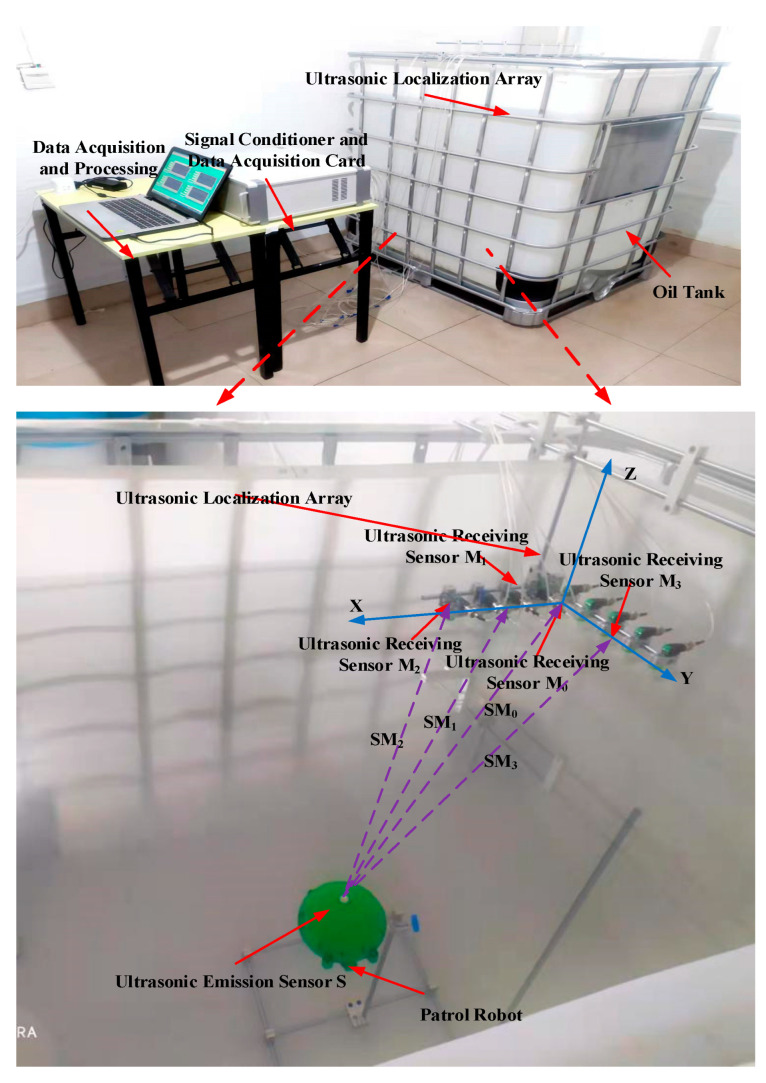
Physical diagram of the 3D spatial positional localization test platform.

**Figure 11 sensors-24-04937-f011:**
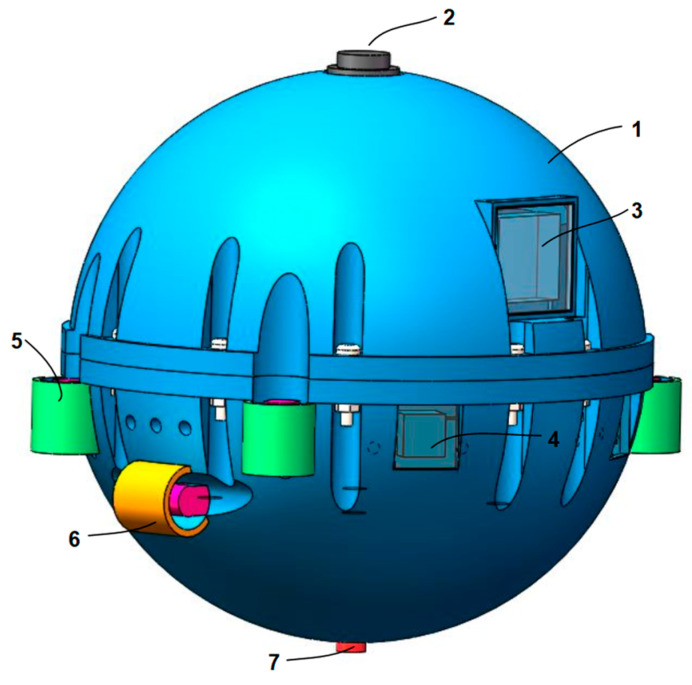
Schematic diagram of transformer internal inspection robot.

**Figure 12 sensors-24-04937-f012:**
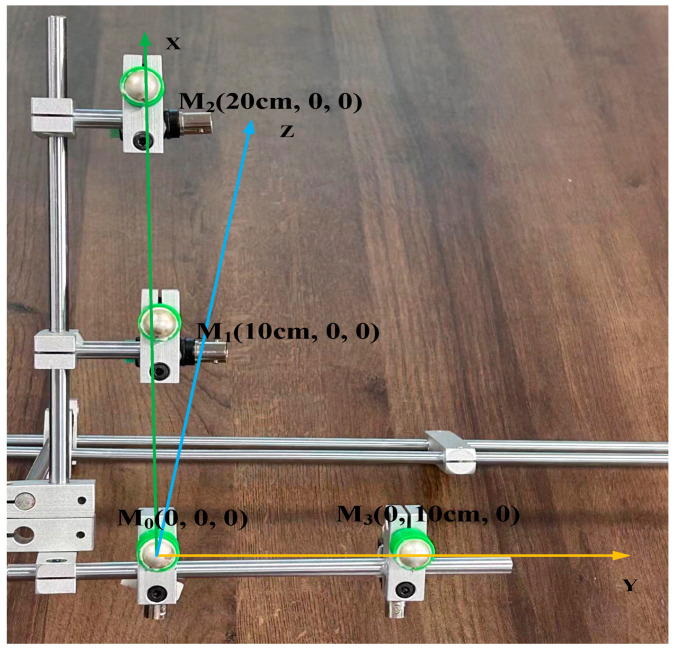
L-shaped sensor array.

**Figure 13 sensors-24-04937-f013:**
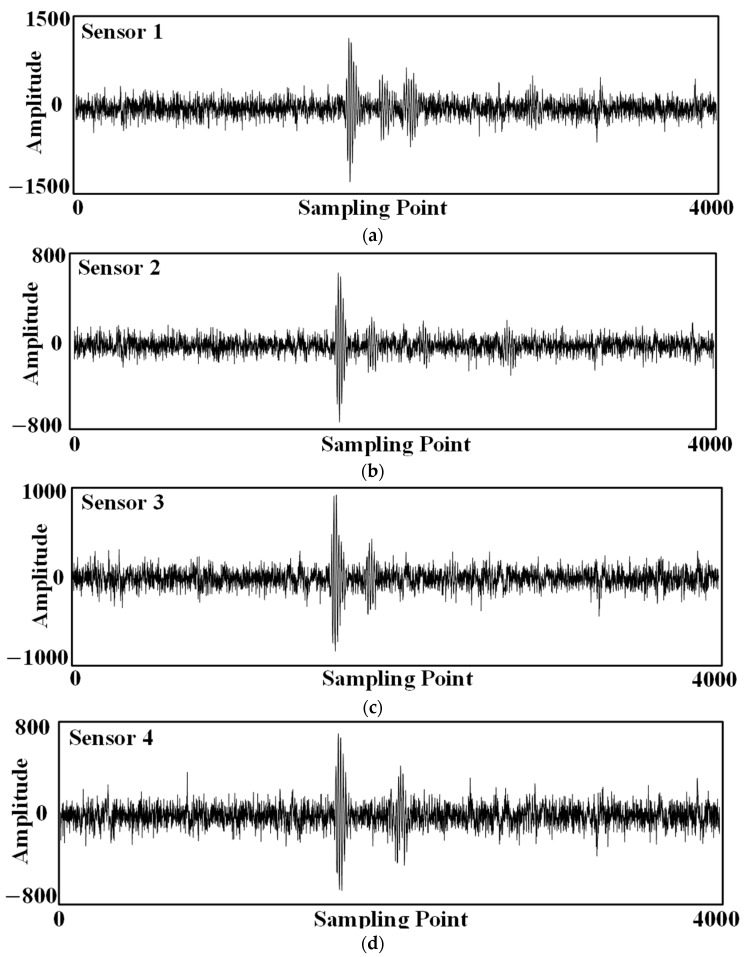
Ultrasound signals received by the ultrasound array: (**a**) Sensor No. 1; (**b**) Sensor No. 2; (**c**) Sensor No. 3; (**d**) Sensor No. 4.

**Figure 14 sensors-24-04937-f014:**
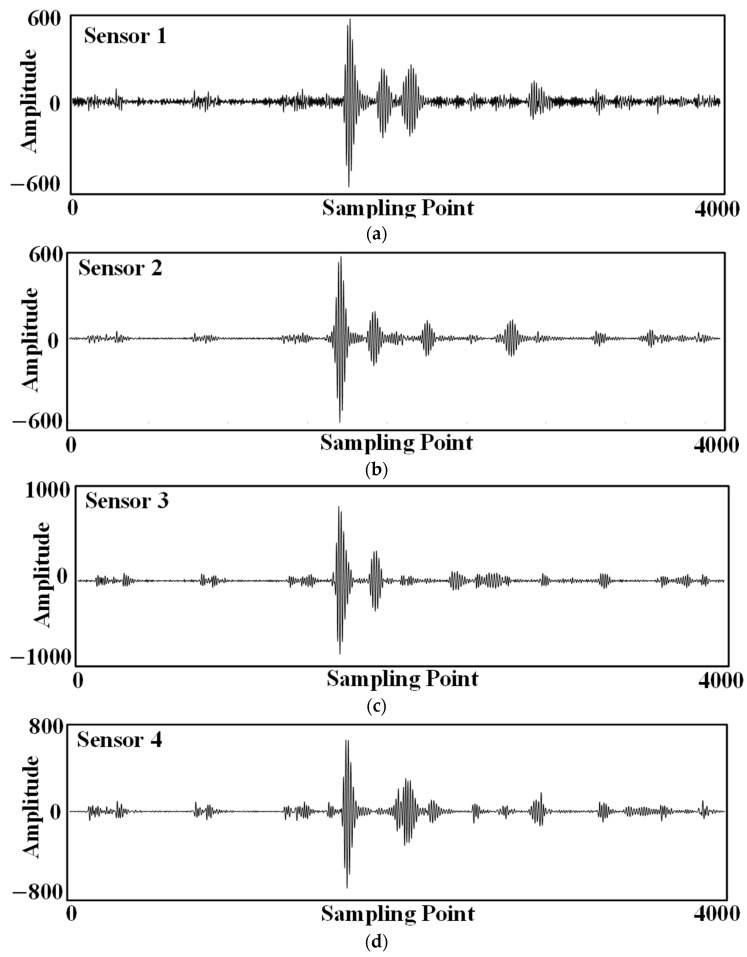
Ultrasonic waveforms after the denoising: (**a**) Sensor No. 1; (**b**) Sensor No. 2; (**c**) Sensor No. 3; (**d**) Sensor No. 4.

**Table 1 sensors-24-04937-t001:** Ultrasound (longitudinal wave) propagation speed.

Medium	Speed m/s
hydrogen	1280
air	330
SF_6_	140
mineral oil	1400
water	1480
porcelain	5600–6200
Oil-paper	1420
steel	6000
copper	4700
epoxy resin	2400–2900
polyethylene	2000

**Table 2 sensors-24-04937-t002:** Comparison of denoising effect evaluation.

Denoising Method	SNR/dB	RMSE	NCC
Traditional EMD	4.2763	0.1009	0.8137
Improved EMD	6.5891	0.0708	0.8966
Bandpass filtering	4.0836	0.0945	0.7874

**Table 3 sensors-24-04937-t003:** Delay results based on SCOT-β.

Num	Delay_20_ (μs)	Delay_10_ (μs)	Delay_30_ (μs)
1	38.4	23.6	10.0
2	38.4	23.6	10.0
3	38.4	23.6	10.0
4	38.8	23.6	10.0
5	38.8	24.0	10.0
6	38.8	23.6	9.6
7	38.4	23.6	10.0
8	38.8	23.6	9.6
9	38.4	24.0	9.6
10	38.8	23.6	10.0
11	38.4	23.6	10.0
12	38.8	24.0	9.6
13	38.4	23.6	9.6
14	38.8	24.0	10.4
15	38.8	23.6	10.4
16	38.4	23.6	9.6
17	38.8	24.0	9.6
18	38.8	24.0	9.6
19	38.8	23.2	10.4
20	38.4	23.6	10.0

**Table 4 sensors-24-04937-t004:** Results of delay estimation and positional localization.

Num	Delay_20_ (μs)	Delay_10_ (μs)	Delay_30_ (μs)	Position Coordinate/cm
1	38.8	23.6	10.0	31, 18, −71
2	38.8	24.0	9.6	30, 20, −70
3	38.4	23.6	9.6	31, 18, −70
4	38.4	24.0	9.6	31, 20, −70
5	38.8	24.0	10.0	31, 19, −71
6	38.8	23.6	9.6	30, 20, −71
7	38.4	23.6	10.0	31, 18, −71
8	38.8	23.2	10.4	32, 20, −73
9	38.8	24.0	10.4	31, 20, −71
10	38.8	23.6	10.4	32, 19, −72

## Data Availability

The data presented in this study are available on request from the corresponding author. The data are not publicly available due to privacy.

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
