# Peer review of "Spatial Localization of Transformer Inspection Robot Based on Adaptive Denoising and SCOT-β Generalized Cross-Correlation"

_sensors, 2024, doi:10.3390/s24154937_

Round 1
Reviewer 1 Report
Comments and Suggestions for Authors
It would be good if there was a list of abbreviations
How is the weighting factor SCOT-ß obtained? empirically or iteratively
Reviewer 2 Report
Comments and Suggestions for Authors
In the manuscript, the authors propose a three-dimensional position localization approach for transformer inspection robot using adaptive denoising and Smoothed Coherence Transform (SCOT) generalized inter-correlation for L-type ultrasonic arrays. The approach is verified via in-situ experiments in which the absolute errors of 0.9 cm, 1.1 cm, and 1.15 cm were obtained in the X, Y, and Z axes, respectively. Authors are invited to respond to the following notes:
1. In the abstract and in the main text, the abbreviations not described are used.
2. The format of the references does not meet the requirements of the journal. Please refer to the Instructions for Authors and review the entire manuscript.
3. The meaning of the symbol used on pages 3 (line 8), 7 (line 5), and 11 (line 19) is unclear.
4. In my opinion, section 2 does not contain any new findings in the field of applied acoustics. I suggest deleting this section.
5. On page 5, the work of Dragomiretskiy et al. is mentioned but not referenced.
6. If a figure consists of numerous sub-figures (e.g. Figure 5), each sub-figure should be identified by a letter and each sub-figure should be explained in the figure caption. . Please refer to the Instructions for Authors and review the entire manuscript.
7. In Figure 7, some subfigures are stretched in height.
8. In conclusion, the authors mention that the obtained results meet the current technical requirements for the position localization of the inspection robot. However, these requirements are not discussed in the paper. What are the limiting values that allow to consider the proposed approach as an efficient technique?
Reviewer 3 Report
Comments and Suggestions for Authors
This paper proposes a three-dimensional positioning method based on adaptive noise reduction and SCOT-β generalized cross-correlation for the positioning of the internal inspection robot of oil-immersed transformers. This method achieves good positioning results in low signal-to-noise ratio environments by improving the EMD noise reduction and generalized cross-correlation algorithms. Experimental results show that this method has potential feasibility in engineering applications. However, there are still some aspects that can be improved in this paper to enhance its scientificity and practicality:
1. The improved EMD method avoids the modal aliasing and endpoint effects of the traditional EMD method. However, the selection criteria for the coefficients K and α need further clarification. Why were the coefficients K set to 5 and α set to 2000 in the paper?
2. In Section 5.1, it is mentioned that 'The ultrasonic array mounted inside the transformer consists of 4 omnidirectional waterproof piezoelectric ceramic ultrasonic probes, with 3 probes in the horizontal direction and 2 probes in the vertical direction, constituting an L-shaped ultrasonic array.' Could you clarify why there are a total of 4 probes in the array, including 3 horizontal and 2 vertical probes?
3. In Section 5.3, are there any relevant standards or requirements that define the 'engineering requirements' for positioning error? Could you provide citations for these standards?
4. The paper assumes that there is only transformer oil inside the oil-immersed transformer and that waves propagate in a straight line. However, in practical applications, numerous internal structures within the transformer can interfere with the wave propagation path, causing refraction and reflection. Has the author considered these issues in actual applications?
Round 2
Reviewer 2 Report
Comments and Suggestions for Authors
The authors did not fully address my comments.
1. Full names of abbreviations in the abstract have been added. However, it is much more common to give the full name followed by the abbreviation in parentheses, not vice versa.
2. References should not be superscripted according to journal requiremen. They should be written in regular text.
3. The reference with the number in the paper to the work of Dragomiretskiy et al. has not yet been provided.
4. The format of figures with multiple subfigures still does not meet the journal's requirements. Carefully consider these requirements and make the changes in the format of these figures.
